# Costs for Statutorily Insured Dental Services in Older Germans 2012–2017

**DOI:** 10.3390/ijerph18126669

**Published:** 2021-06-21

**Authors:** Aleksander Krasowski, Joachim Krois, Sebastian Paris, Adelheid Kuhlmey, Hendrik Meyer-Lueckel, Falk Schwendicke

**Affiliations:** 1Department of Oral Diagnostics, Digital Health and Health Services Research, Charité—Universitätsmedizin Berlin, Aßmannshauser Str. 4-6, 14197 Berlin, Germany; Aleksander.krasowski@charite.de (A.K.); joachim.krois@charite.de (J.K.); 2Department of Restorative, Preventive and Pediatric Dentistry, zmk Bern, University of Bern, 3012 Bern, Switzerland; hendrik.meyer-lueckel@zmk.unibe.ch; 3Department of Operative and Preventive Dentistry, Charité—Universitätsmedizin, 10117 Berlin, Germany; sebastian.paris@charite.de; 4Institute of Medical Sociology and Rehabilitation Science, Charité—Universitätsmedizin, 10117 Berlin, Germany; adelheid.kuhlmey@charite.de

**Keywords:** access, costs, geriatrics, gerodontology, health services research, health economics

## Abstract

Objectives: We assessed the costs of dental services in statutorily insured, very old (geriatric) Germans. Methods: A comprehensive sample of very old (≥75 years) people insured at a large Northeastern statutory insurer was followed over 6 years (2012–2017). We assessed dental services costs for: (1) examination, assessments and advice, (2) operative, (3) surgical, (4) prosthetic, (5) periodontal, (6) preventive and (7) outreach services. Association of utilization with: (1) sex, (2) age, (3) region, (4) social hardship status, (5) International Disease Classification (ICD-10) diagnoses and (6) Diagnoses Related Groups (DRGs) was explored. Results: 404,610 individuals with a mean (standard deviation, SD) age 81.9 (5.4 years) were followed, 173,733 did not survive follow-up. Total mean costs were 129.61 (310.97) euro per capita; the highest costs were for prosthetic (54.40, SD 242.89 euro) and operative services (28.40, SD 68.38 euro), examination/advice (21.15, SD 28.77 euro), prevention (13.31, SD 49.79 euro), surgery (5.91, SD 23.91 euro), outreach (4.81, SD 28.56 euro) and periodontal services (1.64, SD 7.39 euro). The introduction of new fee items for outreach and preventive services between 2012 and 2017 was reflected in costs. Total costs decreased with increasing age, and this was also found for all service blocks except outreach and preventive services. Costs were higher in those with social hardship status, and in Berlin than Brandenburg and Mecklenburg-Western Pomerania. Certain general health conditions were associated with increased or decreased costs. Conclusions: Costs were associated with sex, social hardship status, place of living and general health conditions. Clinical significance: Dental services costs for the elderly in Germany are unequally distributed and, up to a certain age or health status, generated by invasive interventions mainly. Policy makers should incentivize preventive services earlier on and aim to distribute expenses more equally.

## 1. Introduction

There is an increasing focus in health services research to study the effects of high age on medical services costs, mainly as populations are ageing and the elderly are the only growing age group in many high-income countries. Knowledge on costs in this group and drivers of it are useful for health services planning, but also to assist developing targeted interventions for improving health at increased efficiency. Compared with medical costs throughout most of the life course, end-of-life costs have been found to be high, mainly driven by repeated inpatient (hospital) admissions [1]. Costs have been associated with aspects like sex (with ambiguous data finding either male or female individuals experiencing higher medical costs towards the end of life) [2,3,4], the place of living (with costs being higher in more urban and affluent areas) [1,4,5], socio-economic status (SES, with costs being higher in more affluent high-SES groups) [6] or age (with end-of-life costs being consistently higher in younger-dying individuals than those dying older) [2,4,7,8].

While dental costs are only a fraction of overall lifetime medical expenditures [9], costs for dental services in the elderly are of growing relevance. In the United States, for example, per capita costs for dental care have grown faster over the last decade than costs for other healthcare services in the elderly [10], possibly as dental health improvements in children and adults, indicated by fewer restored or missing teeth [11,12], have so far not been fully translated into higher age: While the very old retain a higher number of teeth than ever, they concomitantly suffer from a higher number of coronal and root caries lesions as well as periodontal disease, requiring complex restorative and periodontal interventions as well as different kind of prosthetic management (replacing fewer teeth; for example, edentulism is an increasing rare phenomenon) [13]. Dental costs have been found to be affected by a range of factors, namely age (decreasing in high age) [9,14], sex (mainly via life expectancy differences and the different “chance” to utilize services in higher age) [9], but also insurance type or SES [14,15].

In a recent series of studies, we evaluated the utilization of general, operative, surgical, periodontal and prosthetic dental services in the very old (geriatric), defined as 75 years or older, in Germany [16,17,18]. In the present evaluation, using claims data from one large statutory insurer, we translate utilization into costs. While claims suffer from a range of limitations like selection, confounding or misclassification bias, they do not suffer from reporting bias, are generalizable and reflect the true costs occurring to payers [19,20]. We hypothesized that costs decreased with age, concomitant to a decrease in utilization, but that this decrease was not uniform across different services blocks. Moreover, we hypothesized that individuals’ general health, their SES and place of living are associated with costs.

## 2. Methods

### 2.1. Study Design

The underlying dataset has been used to assess the utilization of general, operative and surgical [16], periodontal [17] and prosthetic [18] dental services by older Germans. The dataset contained claims data from a statutory (public) health insurance in Germany, the AOK Nordost, with a cohort of elderly followed over 6 years (2012 to 2017). The AOK Nordost is the Northeastern branch of a large national insurer, the Allgemeine Ortskrankenkasse (AOK), active mainly in the federal states of Berlin, Brandenburg and Mecklenburg-Western Pomerania. Notably, individuals insured here may have moved into other German states; we excluded these for geographic analyses. The reporting of this study follows the RECORD statement [21].

### 2.2. Setting

Around 1.8 million individuals from the German capital, Berlin, and two rural states, Brandenburg and Mecklenburg-Western Pomerania, with only few larger cities (>70,000 inhabitants) are insured at AOK Nordost. We used claims data, which had been routinely collected and provided under an ethical approval in a pseudonymized form using a data protection cleared platform via the scientific institute of the AOK Nordost, the GEWiNO (Berlin, Germany).

### 2.3. Participants and Sample Size

A comprehensive sample of individuals aged 75 years or above, insured with the AOK Nordost in 2012, was drawn (*n* = 404,610). Individuals were followed over 6 years. No formal sample size estimation was performed. Variable ascertainment was not possible. Database curation had been done by the GEWiNO. 

### 2.4. Variables

Our outcome was the costs for dental services provided to these elderly people. Within the statutory German insurance, dental services are provided on a fee-per-item basis using fee items catalogues of the statutory insurance. In the Northeast, nearly 90% of individuals are statutorily insured, where items are drawn from the fee item catalogue Bewertungsmaßstab (BEMA). For the present study, a comprehensive analysis of all dental services items was performed. Items were grouped into (1) examination, assessment and advice and wider general services, including radiographs, and auxiliary services mainly related to emergency care; (2) operative services, mainly relating to direct restorations (fillings) and endodontic services; (3) surgical services, i.e., extractions, apicectomies and pre-prosthetic surgery; (4) periodontal services, i.e., planning and assessment of periodontal services, and non-surgical and surgical scaling and root-planing; (5) prosthetic services, including crowns (which fall into prosthetic services in German insurance regulations), fixed and removable (partial and total) dentures and repairs, relinings etc.; (6) prevention, including oral hygiene assessment and advice and calculus removal or denture cleaning (these services were increasingly reimbursed since 2013); (7) outreach services, including outreach visits in care homes (these were re-organized in 2013, with new and additional fees being available). Note that implant placement is not covered by the statutory insurance, while coronal restoration is partially covered (these are also covered for tooth-retained crowns), which is why costs for implants are not reflected by our estimates.

Costs were assessed, as previously, according to (1) sex (male/female); (2) age (in years) in each year of follow-up; (3) region; we used municipalities including the capital Berlin (with over 3.5 million inhabitants), medium-sized cities (70,000–200,000 inhabitants), and rural areas; (4) social hardship status (income <1246 euro/month per capita in 2019); (5) International Disease Classification (ICD-10) diagnoses, derived from outpatient diagnostic data; (6) inpatient hospital diagnoses and treatments, derived from German-Modified Diagnoses Related Groups (DRGs). The DRGs classify diseases in groups of similar pathogenesis, characteristics and treatment complexity. Only the 25 most frequently recorded ICD-10 and DRG codes were used.

### 2.5. Data Sources and Access

Data were provided by the GEWiNO using a data protection approved platform. Data were pseudonymized and included the described covariates and BEMA items claimed per year. Comparability of data between different years and data consistency was given within the limitations outlined (few items were newly introduced, changed or terminated during the period 2012–2017).

### 2.6. Bias

Participants and providers were not aware that any routine data analyses would be performed with collected claims data later on; the risk of bias in this domain is hence low. Selection bias for the target population (very old individuals at AOK Nordost) was not possible, while it should be emphasized that the overall population of very old Germans differs from our sample, as discussed in detail later on.

### 2.7. Statistical Analyses

Descriptive statistics of age groups were computed based on the age distribution in 2012. An individual was assigned to having a social hardship if the individual was assigned to this status at least once during the period 2012 to 2017. For geographical analysis we excluded all individuals that relocated from one of the federal states (Berlin, Brandenburg and Mecklenburg-Vorpommern) to another federal state, which decreased the sample size to 390,044. We did not correct for relocations within the three federal states. For ICD-10 and DRG codes, we summed up all claims, selected the 25 most frequent diagnoses in each group (in total 50) and computed the number of individuals who were assigned to having this diagnosis or treatment 2012 to 2017. 

We applied ridge regression, a regularized form of linear regression that is less affected by multicollinearity and less prone to overfitting [22]. The response variable was costs. As predictor variables we included age, gender, being deceased, social hardship status, federal state (note that we allowed the category ‘other’ for relocated individuals) and the described outpatient and inpatient hospital diagnosis variables referring to the year 2014 (as the BEMA items by large remained consistent from this year onwards). Confidence intervals for covariates were established applying bootstrapping. All analyses, modeling and visualization were performed using Python (version 3.7) and auxiliary modules (http://www.python.org, accessed on 26 May 2021).

## 3. Results

A total of 404,610 individuals were followed over up to 6 years (Table 1); 173,733 did not survive follow-up. The mean follow-up was 1689 days (standard deviation SD: 705). The mean (SD) age of the sample was 81.9 (5.4) years. The majority of individuals were female and younger than 85 years. Nearly half of them claimed hardship status once during the observational period. 

Total mean costs were 129.61 (310.97) euro per capita; the highest costs were for prosthetic services (54.4, SD 242.89 euro), followed by operative services (28.40, SD 68.38 euro) and examination/advice (21.15, SD 28.77 euro), prevention (13.31, SD 49.79 euro), surgical (5.91, SD 23.91 euro), outreach (4.81, SD 28.56 euro) and periodontal services (1.64, SD 7.39 euro). Costs were slightly higher in males than females, with the notable exception being costs for outreach services. Total mean costs decreased with increasing age, and this was also found for all service blocks except outreach and preventive services. Total mean costs were higher in those with social hardship status, mainly due to increased costs for outreach and preventive services. Total mean costs were higher in Berlin than Brandenburg and Mecklenburg-Vorpommern, and in particular prosthetic, prevention and outreach costs were higher in Berlin.

Costs decreased with each year of age until age 95 years; this pattern was consistent over time (Figure 1). Notably, in 2012 costs decreased further beyond this age, while in 2013 and more so the subsequent years, costs stabilized from 95 years onwards. This was mainly due to cost decreases in most services blocks (examination/advice, operative, prosthetic) being compensated by increased costs for outreach services and prevention. Costs for prosthetic services were higher in 2012 than the subsequent years in all age groups, while for operative, surgical and periodontal costs a moderate increase over the years was observed. The increase was more pronounced for outreach services, and drastic for preventive costs: Here, a multifold increase, especially in higher age, was observed.

Costs were higher in larger cities (in Berlin, they exceeded a mean of 150 euro, while in some rural areas they were below 100 euro) (Figure 2). This was mainly due to costs for examination/advice and prosthetic services, while those for operative, periodontal and outreach services showed a less obvious pattern. Costs were generally highest in Berlin, regardless of which services were assessed.

Costs according to ICD-10 codes (Table 2) were higher for the majority of codes, e.g., eye conditions (e.g., presbyopia, cataract, astigmatism), gonathrosis, cox-arthrosis, osteoporosis and unspecified chronic pain. Prevention costs stayed close to the global mean for most codes, whereas outreach costs were usually lower. Notably individuals with dementia, heart failure and urinary incontinence had higher prevention and outreach costs, but lower costs in remaining categories. An opposite pattern was observed for individuals with eye conditions, for example.

We further assessed the costs stratified according to different DRGs (Table 3). Costs were highest in patients hospitalized for non-complicated cardial diagnostics, non-surgically treated diseases and injuries of the spinal column, joint operations, geriatric complex treatment and eye operations. These also came with relatively low outreach and prevention costs. Costs were lowest for serious cardiac insufficiency, paraplegia/tetraplegia, renal insufficiency and geriatric rehabilitation patients. Outreach and prevention costs were higher for the majority of codes. 

In multivariable analysis (Table 4), mean total costs (in 2014) were significantly lower with each year of age, in those deceased during follow-up, and in Mecklenburg-Western Pomerania or Brandenburg than Berlin, and higher in males than females and those with social hardship status. Nearly all general health conditions were also significantly associated with costs.

## 4. Discussion

The costs for general healthcare in the elderly has been found to increase with proximity to death, and has been associated with sex [2,3,4], place of living [1,4,5], SES [6] or age, as described [2,4,7,8]. In the present evaluation, we assessed costs for dental services provided to the very old by one statutory insurer in Germany. We found total costs to decrease with age, while the costs for specific services did not necessarily follow that pattern and were affected by policy measures introduced during the period 2012–2017. Moreover, we found costs to be by and large generated by operative or prosthetic interventions, and to be unequally distributed. Younger and healthier individuals and those in urban regions as well as individualized with a social hardship status (i.e., low SES) had higher costs. The latter finding is particularly notable and will be discussed later on in detail. 

Generally, our findings need in-depth exploration. First, we found total costs to decrease with age, but to plateau at a low level in those aged 95 years onwards in 2013 and the subsequent years. We also found costs to be minimally higher in males than females (except for outreach services). The cost decrease can be expected with older individuals who have a higher morbidity showing lower mobility. Dental services may also be down-prioritized in these individuals, with accessing other healthcare domains being of higher relevance. The observed effect of a plateau of costs at very high age was due to a shift in services pattern; cost decreases for prosthetic and operative services were compensated by increases for outreach and preventive care; from 2015 onwards, these even over-compensated the costs for invasive services in this age group, reflecting the introduction and adoption of new fee items. Notably, the main costs for those aged 75–95 years were nevertheless prosthetic and operative costs. This is worrisome, as dentists seem to perpetuate an invasive servicing model up to the point where this is not viable any longer given patients’ high age and sickness, while preventive services are not provided until they may be considered as a “last resort”. However, this is determined by remuneration policies in Germany; preventive services are only provided to those elderly requiring care assistance (Pflegestufe). Policy makers should consider incentivizing prevention already in the “younger” elderly and thereby drive servicing towards sustainability.

The cost difference between sexes was limited. Service consumption differences will have driven these cost differences [16,17,18], but also differences in confounders. For example, females were generally older (as they survived longer) and showed a higher level of care requirements in this older age. While we cannot fully ascertain the reasons for the observed (limited) cost differences, we showed differences in utilization along sexes in a similar direction before, indicating that both health status but also service consumption patterns may differ [16,17,18].

Second, we assessed the impact of social hardship status on costs. Hardship status is a proxy for low income; individuals with this status receive prosthetic care at nearly no out-of-pocket costs given increased reimbursement rates by the statutory insurance to the provider. We have found this status to increase prosthetic service utilization in the very old before [18], and now confirm that generally, expenses (as a proxy for utilization) are increased in this group (note that for prosthetic services, this effect is aggravated as the insurance costs per prosthetic case are increased given higher per capita reimbursement). While we do not know if servicing was appropriate and suited to improve health, it is reassuring to see that at least in this group the incentives towards equitable accessibility and service provision are effective given that those with low social status are also likely to show the poorest oral and general health [23].

Third, and in line with previous examinations, we found pronounced differences in utilization according to place of living [24]. This may be due to higher dentists’ densities in urban areas, leading to increased accessibility, shorter travel times for dentists providing outreach services, or supply-side-induced demand [25,26,27]. Given the expected decreases in the dental workforce in more rural areas in the next decade [27,28], it can be expected that the regionally unequal servicing of the very old will increase. Policy makers are called upon to incentivize servicing in rural areas and to reflect the additional efforts and costs for providing services to the very old, especially those living in long-term care, in rural instead of urban areas.

Fourth, we found costs to differ according to health status. Those consuming specific services related to non-serious conditions, like those of the eye or of the joints, usually generated higher costs, reflecting a different utilization pattern than those with life threatening conditions. Based on our cost estimates, we confirm the idea of different strata in the elderly, colloquially termed as “go-goes, slow-goes, no-goes”. Our findings highlight that for the last group, the very sick, in long-term care or long-term hospitalization, dental services are by large inaccessible. Given the relevance of oral health for wider health, dental research and dental interest groups should develop policies of providing services to these individuals, too [29]. This should be done especially in consideration of the high efforts associated with providing such services and the limited remuneration in Germany at present.

This study has a number of strengths and limitations. First, it is the first longitudinal investigations evaluating dental services costs in the very old. Our cohort, with over 400,000 individuals from three different federal states, is large and likely representative for the target population of statutorily insured in the Northeast of Germany. Second, a wide range of factors associated with costs were assessed, e.g., ICD-10 or DRG codes, providing new insights which we could not discuss in depth here. Third, and as a limitation, claims data suffer from a range of biases. Causality, inference with other relevant factors (e.g., medication, care assistance status) or with needs and health effects cannot be deduced from our data. Last, individuals insured by AOK Nordost are not representative for all Germans; they are usually less affluent and older than the national average. Moreover, the Northeast shows particular rural–urban disparities, with Berlin as capital and some of the poorest and least inhabited areas of Germany in direct proximity. Our sample does also not allow any insights into costs for out-of-pocket expenses or privately insured individuals (their number is limited in this area of Germany and overall, around 11% of all Germany). Overall, it could be speculated that similar socio-demographic aspects may apply to other populations in Germany, too, at least those covered by the statutory insurance, while the magnitude of differences between population subgroups may differ. For patients who pay privately, costs and cost differences will differ, as cost estimation is different, but also as these individuals can be expected to be healthier, more mobile, but often need to pay for dental services upfront out-of-their-pocket, all of which will determine service utilization and costs.

When summarizing possible policy angles, based on our data and the outlined limitations: (1) policy-makers and payers can expect a decrease in costs for dental services to some degree if morbidity and utilization patterns remain the same or morbidity decreases in the very old, while this group ages and the subgroup of very old grows. It is, however, conceivable that the very old of tomorrow will be differently sick, mobile and require different dental treatments. (2) Costs differ between socio-demographic groups; policies should be derived to allow more equal consumption of services (generating equal costs). (3) Costs as a result of utilization also differed along health groups. Servicing concepts for dental services in the very sick (i.e., less mobile ones) and those living remote, and particularly those very old and requiring care assistance are needed. The fact that outreach costs have increased are assuring, demonstrating that servicing is indeed steered towards where needs are (driven by policy measures of the past). Notably, however, outreach services seem to continuously provide services which are rather symptomatic and not sustainable on their own (e.g., restorative and prosthetic care). A more preventive and intersectoral approach is needed. (4) The impact of the recent coronavirus disease 2019 (COVID-19) pandemic and the associated lockdowns of most care homes as well as the massively aggravated limitations in mobility of the very old are not reflected in this study. Future studies should compare cost estimates from that period with ours, which can serve as a pre-COVID baseline to some degree. 

## 5. Conclusions

In conclusion, costs for dental services in the very old in Northeast Germany were associated with sex, place of living, social hardship status and general health conditions. Moreover, up to a certain age or health status, costs were generated by invasive interventions mainly. Policy makers should incentivize preventive services earlier on and aim to distribute expenses more equally.

## Figures and Tables

**Figure 1 ijerph-18-06669-f001:**
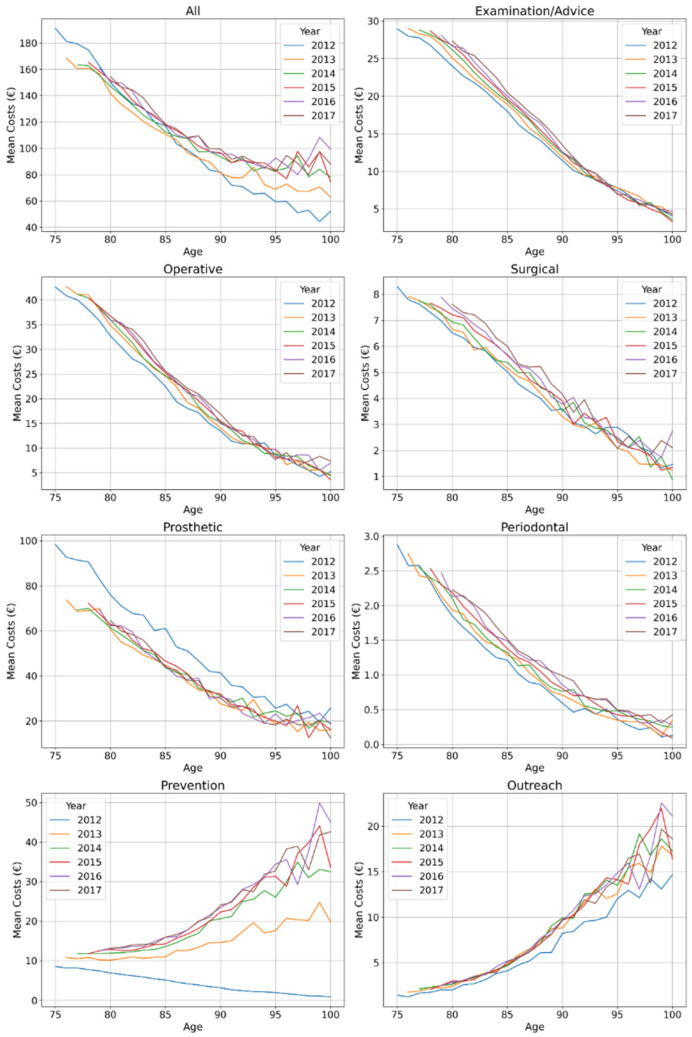
Mean costs (in euro 2012) of dental services provided to the very old in Northeast Germany. Total costs as well as costs for examination/advice, operative, surgical, prosthetic, periodontal, preventive and outreach services are shown. Individuals available in 2012 of all ages from 75 years upwards (blue line) were followed over 6 years until 2017 (brown line), i.e., the 75-year-olds in 2012 are the 76-year-olds in 2013 etc. (which is why the lines start further to the right with longer follow-up).

**Figure 2 ijerph-18-06669-f002:**
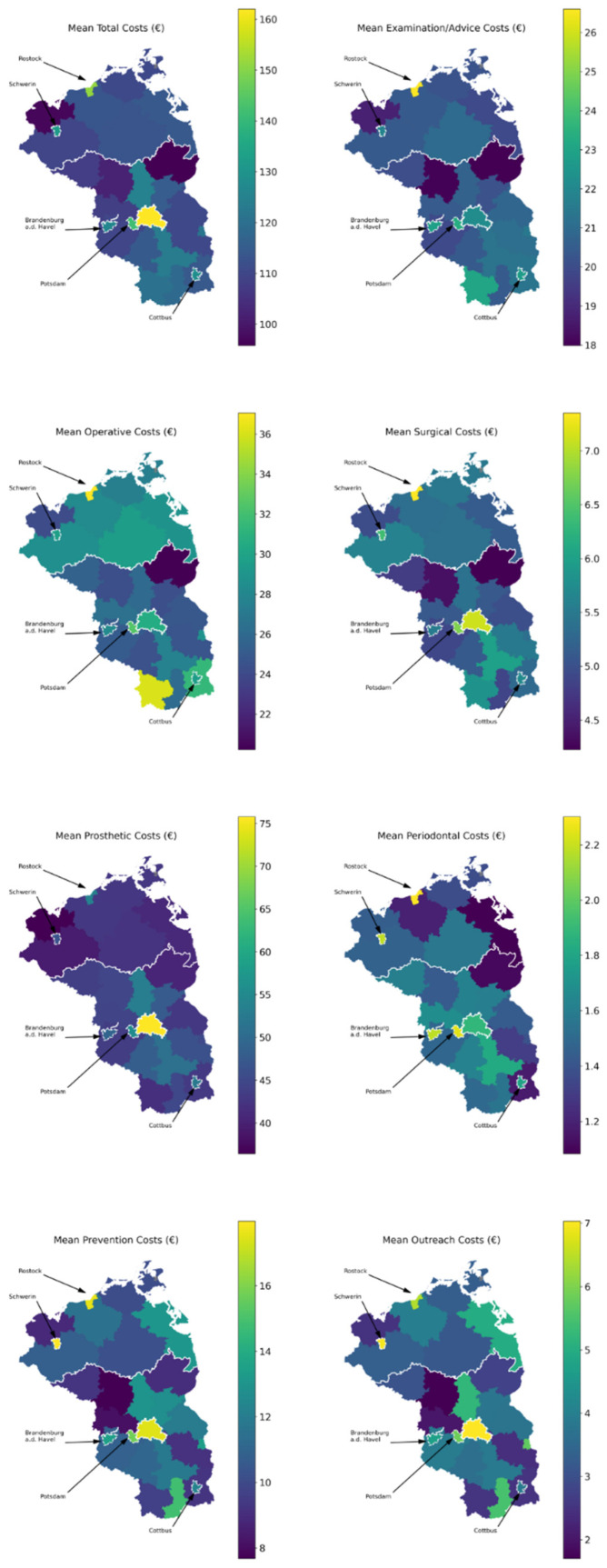
Regionally specific mean costs across the years (in euro 2012) of dental services provided to the very old in Northeast Germany. Total costs as well as costs for examination/advice, operative, surgical, prosthetic, periodontal, preventive and outreach services are shown. Larger cities are highlighted by arrows.

**Table 1 ijerph-18-06669-t001:** Sample characteristics (N, %) and mean costs (± standard deviation (SD)) for dental services for the very old in Northeast Germany. Total, male and female population aged 75 years or older, in 5-years age bands, according to social hardship status and federal state are shown.

Covariate		Total [N (%)]	All	Examination/Advice	Operative	Surgical	Prosthetic	Periodontal	Prevention	Outreach
Total		404,610 (100.0%)	129.61 (±310.97)	21.15 (±28.77)	28.4 (±68.38)	5.91 (±23.91)	54.4 (±242.89)	1.64 (±7.39)	13.31 (±49.79)	4.81 (±28.56)
Sex	male	134,909 (33.34%)	140.52 (±322.51)	24.01 (±30.54)	33.75 (±76.64)	7.11 (±26.83)	58.77 (±252.92)	1.84 (±7.93)	11.67 (±42.54)	3.37 (±24.48)
	female	269,701 (66.66%)	124.29 (±305.05)	19.76 (±27.77)	25.79 (±63.81)	5.33 (±22.33)	52.27 (±237.83)	1.54 (±7.11)	14.1 (±52.94)	5.51 (±30.32)
Age group	75–79	162,368 (22.75%)	167.11 (±359.4)	27.85 (±32.15)	39.9 (±82.2)	7.61 (±27.3)	77.07 (±292.29)	2.45(±10.17)	10.23 (±29.51)	2.0 (±19.22)
	80–84	266,956 (37.4%)	136.81 (±317.93)	23.63 (±29.77)	31.64 (±71.68)	6.53 (±25.14)	58.37 (±252.78)	1.8 (±7.49)	11.57 (±40.8)	3.27 (±23.21)
	85–89	174,673 (24.47%)	106.39 (±275.03)	17.06 (±25.41)	21.35 (±57.29)	4.88 (±21.44)	41.47 (±206.55)	1.15 (±5.21)	14.35 (±57.65)	6.11 (±32.42)
	90–94	82,597 (11.57%)	86.73 (±247.58)	10.63 (±20.1)	12.78 (±41.88)	3.33 (±17.65)	28.23 (±166.88)	0.66 (±3.43)	20.29 (±75.27)	10.81 (±42.58)
	95–99	22,641 (3.17%)	79.61 (±232.68)	6.47 (±15.57)	7.76 (±29.86)	2.15 (±13.46)	20.71 (±139.91)	0.38 (±2.29)	27.08 (±85.58)	15.05 (±46.71)
	100–104	4214 (0.59%)	65.67 (±195.15)	3.47 (±10.61)	4.58 (±23.26)	1.06 (±8.75)	12.41 (±100.0)	0.18 (±1.81)	27.4 (±81.52)	16.56 (±46.05)
	105–109	348 (0.05%)	56.54 (±158.57)	2.48 (±8.63)	2.51 (±11.61)	0.74 (±7.47)	6.51 (±64.99)	0.18 (±1.43)	29.21 (±78.34)	14.92 (±37.79)
Hardship status	no	210,292 (51.97%)	123.11 (±303.12)	21.0 (±28.74)	27.85 (±67.39)	5.79 (±23.52)	52.69 (±239.88)	1.75 (±7.79)	10.85 (±39.32)	3.18 (±22.65)
	yes	194,318 (48.03%)	136.47 (±318.91)	21.31 (±28.82)	28.98 (±69.41)	6.05 (±24.32)	56.2 (±246.02)	1.52 (±6.93)	15.9 (±58.75)	6.52 (±33.61)
Federal state	Berlin	122,454 (30.26%)	162.02 (±375.42)	22.17 (±30.66)	30.61 (±70.15)	7.12 (±27.06)	75.87 (±293.86)	1.89 (±7.85)	17.37 (±65.43)	6.99 (±37.36)
	Brandenburg	153,164 (37.85%)	114.82 (±277.92)	20.92 (±27.48)	26.77 (±64.71)	5.24 (±21.44)	46.37 (±220.79)	1.57 (±7.18)	10.62 (±36.33)	3.33 (±22.24)
	Mecklenburg-Vorpommern	107,665 (26.61%)	115.19 (±271.93)	21.03 (±28.52)	29.2 (±72.39)	5.58 (±23.42)	42.52 (±208.13)	1.49 (±7.24)	11.5 (±41.56)	3.87 (±22.86)
	Relocated individuals	21,327 (5.27%)	123.93 (±301.18)	17.62 (±27.71)	23.55 (±62.35)	5.58 (±23.88)	49.69 (±228.31)	1.39 (±6.75)	18.42 (±64.16)	7.69 (±35.17)

**Table 2 ijerph-18-06669-t002:** Costs for dental services for the very old in Northeast Germany according to International Disease Classification (ICD-10, German Modification) codes. All and service-specific mean (±SD) costs are shown for the total population and the population with specific ICD-10 codes.

ICD	All	Examination/Advice	Operative	Surgical	Prosthetic	Periodontal	Prevention	Outreach	Description
Total	129.61 (±310.97)	21.15 (±28.77)	28.4 (±68.38)	5.91 (±23.91)	54.4 (±242.89)	1.64 (±7.39)	13.31 (±49.79)	4.81 (±28.56)	-
N40	155.61 (±335.07)	27.07 (±31.76)	38.49 (±81.27)	7.83 (±28.08)	64.12 (±263.35)	2.05 (±8.28)	12.69 (±44.04)	3.38 (±24.71)	Diseases of the genitourinary system
H52.0	154.11 (±337.91)	26.53 (±30.88)	35.81 (±75.98)	7.12 (±26.22)	66.55 (±269.46)	2.08 (±8.46)	12.76 (±44.78)	3.26 (±24.6)	Diseases of the eye and adnex
H52.2	153.49 (±338.32)	26.11 (±30.81)	35.25 (±75.53)	7.09 (±26.18)	66.34 (±268.74)	2.04 (±8.32)	13.11 (±47.02)	3.55 (±25.95)	Diseases of the eye and adnex
H52.4	153.1 (±336.7)	25.99 (±30.65)	35.18 (±75.44)	7.03 (±26.07)	65.43 (±266.52)	2.03 (±8.27)	13.58 (±48.4)	3.86 (±26.93)	Diseases of the eye and adnex
R52.2	150.99 (±342.51)	23.19 (±30.41)	31.8 (±71.88)	6.69 (±25.71)	64.4 (±265.51)	1.75 (±7.47)	16.75 (±61.66)	6.42 (±34.73)	Symptoms, signs and abnormal clinical and laboratory findings, not elsewhere classified
Z96.1	149.95 (±334.54)	25.25 (±30.33)	33.56 (±73.36)	6.8 (±25.65)	64.4 (±263.95)	1.93 (±7.94)	13.79 (±50.92)	4.23 (±28.65)	Factors influencing health status and contact with health services
H26.9	148.14 (±330.89)	25.19 (±30.32)	33.85 (±73.84)	6.83 (±25.69)	63.06 (±261.44)	1.96 (±8.13)	13.36 (±48.33)	3.89 (±26.64)	Diseases of the eye and adnex
E78.0	147.05 (±332.59)	24.79 (±30.33)	33.02 (±72.71)	6.72 (±25.56)	63.79 (±264.52)	2.0 (±8.31)	13.07 (±46.34)	3.66 (±25.42)	Endocrine, nutritional and metabolic diseases
I83.9	146.74 (±330.41)	24.76 (±30.62)	33.37 (±73.28)	6.55 (±25.0)	62.65 (±261.55)	1.89 (±7.88)	13.44 (±48.14)	4.08 (±26.68)	Diseases of the circulatory system
M17.9	146.0 (±331.95)	23.88 (±30.11)	32.33 (±72.29)	6.52 (±25.13)	62.12 (±260.27)	1.82 (±7.71)	14.45 (±53.28)	4.88 (±30.06)	Diseases of the musculoskeletal system and connective tissue
M16.9	145.29 (±328.59)	24.0 (±30.07)	32.66 (±73.09)	6.55 (±25.09)	61.1 (±257.61)	1.8 (±7.56)	14.35 (±52.01)	4.84 (±29.35)	Diseases of the musculoskeletal system and connective tissue
M81.99	144.96 (±327.62)	23.6 (±30.09)	31.73 (±71.17)	6.32 (±24.71)	60.57 (±254.99)	1.76 (±7.47)	15.39 (±56.46)	5.58 (±31.85)	Diseases of the musculoskeletal system and connective tissue
I10.00	142.22 (±322.99)	23.96 (±29.88)	32.11 (±72.02)	6.46 (±24.97)	59.59 (±254.38)	1.83 (±7.75)	13.85 (±49.9)	4.43 (±27.94)	Diseases of the circulatory system
Z92.1	141.05 (±321.03)	24.0 (±30.23)	32.42 (±73.1)	6.65 (±25.76)	58.89 (±251.63)	1.76 (±7.48)	13.16 (±48.9)	4.18 (±27.62)	Factors influencing health status and contact with health services
E78.5	140.44 (±320.67)	23.95 (±29.93)	32.16 (±72.45)	6.47 (±25.01)	59.34 (±253.76)	1.86 (±7.99)	12.83 (±45.62)	3.83 (±25.45)	Endocrine, nutritional and metabolic diseases
I70.9	138.01 (±315.49)	23.25 (±29.67)	30.98 (±71.3)	6.31 (±24.68)	57.18 (±247.43)	1.74 (±7.49)	13.9 (±49.76)	4.65 (±27.67)	Diseases of the circulatory system
E79.0	137.45 (±325.53)	22.22 (±29.18)	29.86 (±70.47)	6.43 (±25.1)	59.32 (±255.82)	1.67 (±7.29)	13.4 (±51.1)	4.55 (±28.63)	Endocrine, nutritional and metabolic diseases
Z25.1	135.98 (±314.53)	22.42 (±29.01)	29.97 (±69.73)	6.1 (±24.2)	55.72 (±245.1)	1.72 (±7.57)	14.75 (±52.93)	5.31 (±29.92)	Factors influencing health status and contact with health services
UUU	132.85 (±314.56)	21.78 (±29.02)	29.25 (±69.26)	6.07 (±24.23)	55.81 (±246.06)	1.69 (±7.5)	13.52 (±50.14)	4.74 (±28.55)	-
I10.90	132.49 (±314.22)	21.74 (±29.0)	29.17 (±69.19)	6.06 (±24.25)	55.6 (±245.74)	1.68 (±7.48)	13.49 (±50.22)	4.75 (±28.62)	Diseases of the circulatory system
I25.9	132.42 (±311.81)	22.12 (±29.33)	29.55 (±69.8)	6.1 (±24.47)	55.03 (±243.54)	1.62 (±7.2)	13.3 (±49.43)	4.71 (±28.37)	Diseases of the circulatory system
E11.90	127.17 (±309.94)	20.55 (±28.11)	27.11 (±66.95)	5.81 (±23.65)	54.04 (±242.79)	1.55 (±7.17)	13.23 (±50.29)	4.88 (±28.92)	Endocrine, nutritional and metabolic diseases
I50.9	125.86 (±310.65)	18.95 (±27.69)	25.4 (±65.03)	5.59 (±23.48)	51.81 (±236.36)	1.33 (±6.24)	15.9 (±62.0)	6.87 (±35.36)	Diseases of the circulatory system
R32	125.1 (±308.05)	17.36 (±26.69)	23.4 (±61.95)	5.33 (±22.91)	49.88 (±230.2)	1.22 (±5.96)	19.02 (±68.73)	8.88 (±38.8)	Symptoms, signs and abnormal clinical and laboratory findings, not elsewhere classified
F03	119.51 (±300.25)	15.14 (±25.43)	20.61 (±58.69)	4.97 (±22.14)	45.32 (±217.89)	1.06 (±5.47)	21.59 (±74.91)	10.83 (±42.17)	Mental and behavioural disorders

**Table 3 ijerph-18-06669-t003:** Costs for dental services for the very old in Northeast Germany according to German Modified Diagnosis Related Groups (DRGs). All and service-specific mean (±SD) costs are shown for the total population and the population with specific DRGs.

DRG	All	Examination/Advice	Operative	Surgical	Prosthetic	Periodontal	Prevention	Outreach	Description
Total	129.61 (±310.97)	21.15 (±28.77)	28.4 (±68.38)	5.91 (±23.91)	54.4 (±242.89)	1.64 (±7.39)	13.31 (±49.79)	4.81 (±28.56)	-
F49G	172.96 (±358.87)	30.93 (±33.16)	42.98 (±84.05)	8.3 (±28.42)	77.08 (±292.91)	2.45 (±9.45)	10.06 (±24.87)	1.16 (±14.19)	Invasive cardiological diagnosis except in acute myocardial infarction, without extremely severe complication or comorbidity, age >17 years, without cardiac mapping, without severe complication or comorbidity at day of treatment> 1, without complex diagnosis, without specific intervention
I68D	149.86 (±333.31)	24.96 (±30.95)	34.82 (±76.06)	6.89 (±26.01)	62.34 (±258.29)	1.76 (±7.4)	14.48 (±55.76)	4.61 (±30.54)	Non-surgically treated diseases and injuries of the spinal column, more than one occupancy day or other femoral fracture, without sacrum fracture, without certain moderately elaborate, elaborate or highly elaborate treatment
I47B	148.4 (±328.7)	24.14 (±30.1)	33.67 (±76.24)	6.67 (±25.82)	60.9 (±254.37)	1.85 (±7.59)	15.84 (±54.28)	5.33 (±30.1)	Revision or replacement of the hip joint without certain complicated factors, with complex diagnosis of the pelvis/thigh, with certain endoprosthetic or joint plastic surgery of the hip joint, with implantation or replacement of a radius head prosthesis.
A90A	146.72 (±333.15)	23.29 (±29.7)	31.35 (±69.88)	6.88 (±25.68)	65.15 (±261.7)	1.7 (±6.82)	13.84 (±53.73)	4.5 (±29.37)	Partial stationary geriatric complex treatment
C08B	145.65 (±337.34)	22.69 (±29.44)	30.21 (±69.59)	6.5 (±24.94)	62.99 (±263.05)	1.66 (±6.88)	15.93 (±60.95)	5.67 (±33.88)	Extracapsular extraction of the lens without congenital malformation of the lens or certain interventions on the lens
F71B	143.77 (±321.77)	24.65 (±30.56)	33.52 (±74.32)	6.8 (±25.71)	60.41 (±252.7)	1.84 (±7.63)	12.84 (±46.67)	3.71 (±25.64)	Non-severe cardiac arrhythmias and conduction disturbances without extremely severe complications or comorbidity or occupancy day, without catheter-assisted electrophysiological examination of the heart, without specific high-level treatment
I41Z	136.29 (±332.54)	18.45 (±27.48)	25.04 (±63.4)	6.01 (±24.56)	58.06 (±250.03)	1.32 (±5.8)	19.11 (±72.48)	8.29 (±40.08)	Geriatric early rehabilitative complex treatment for diseases and disorders of the musculoskeletal system and connective tissue
G67C	136.1 (±321.68)	20.84 (±29.38)	28.25 (±68.31)	6.02 (±24.59)	56.16 (±246.9)	1.38 (±6.18)	16.59 (±62.21)	6.85 (±34.69)	Esophagitis, gastroenteritis, gastrointestinal hemorrhage, ulcer disease and various diseases of the digestive organs without certain or other complicating factors, without extremely severe complications or comorbidity
F67D	133.96 (±313.32)	23.48 (±30.26)	31.89 (±72.53)	6.27 (±24.48)	54.57 (±246.3)	1.68 (±7.21)	12.47 (±47.01)	3.59 (±25.9)	Hypertension without complicated diagnosis, without extremely severe or severe complications or comorbidity, without certain moderately complex/complicated treatment, age >17 years
G67B	132.48 (±313.8)	20.05 (±28.93)	27.32 (±67.02)	5.91 (±23.96)	53.24 (±237.43)	1.39 (±6.29)	17.15 (±63.91)	7.42 (±36.51)	Esophagitis, gastroenteritis, gastrointestinal bleeding, ulcer disease and various diseases of the digestive organs with other complicating factors or with extremely severe complications or comorbidity
B80Z	132.06 (±316.42)	17.56 (±27.89)	24.36 (±63.97)	5.67 (±23.55)	51.18 (±233.2)	1.19 (±5.79)	21.95 (±74.68)	10.16 (±41.28)	Other head injuries
I34Z	127.45 (±318.52)	17.58 (±26.43)	23.79 (±61.7)	5.5 (±23.1)	53.18 (±237.81)	1.27 (±5.93)	18.24 (±70.91)	7.9 (±39.25)	Geriatric early rehabilitative complex treatment with specific operating room procedure for diseases and disorders of the musculoskeletal system and connective tissue
L64A	126.45 (±305.99)	19.07 (±27.89)	25.68 (±64.38)	5.82 (±25.31)	51.12 (±231.69)	1.42 (±6.72)	16.37 (±61.9)	6.96 (±34.23)	Other urinary organ diseases with extremely severe or severe complications or comorbidity or certain diagnosis, more than one occupancy day or urethra-cystoscopy, congenital malformation or age <3 years
B70B	126.04 (±307.52)	18.0 (±26.71)	24.29 (±61.94)	5.5 (±23.1)	50.25 (±232.6)	1.36 (±6.31)	18.51 (±66.83)	8.13 (±38.33)	Apoplexy with complex neurological treatment of acute stroke
J65Z	124.78 (±308.98)	16.92 (±26.35)	22.92 (±61.57)	5.41 (±23.17)	48.53 (±227.62)	1.15 (±5.73)	20.39 (±73.07)	9.47 (±40.42)	Injury of the skin, subcutis and mamma
F73Z	124.41 (±298.86)	19.75 (±28.45)	26.62 (±67.29)	5.76 (±23.62)	49.28 (±227.19)	1.33 (±6.18)	15.55 (±57.91)	6.13 (±31.54)	Syncope and collapse
L63F	124.25 (±307.4)	16.81 (±27.01)	23.11 (±62.58)	5.48 (±23.65)	48.44 (±225.11)	1.13 (±5.65)	19.78 (±70.89)	9.5 (±40.23)	Infections of the urinary organs without extremely severe complications or comorbidity, without certain moderately costly/elaborate/highly costly treatment, without complex treatment multi-resistant pathogens (MRE), without certain serious infections, age >5 and <18 years, without severe complications or comorbidity or age >17 and <90 years
E77I	121.26 (±310.46)	15.86 (±26.02)	21.38 (±60.51)	5.23 (±23.73)	48.2 (±229.64)	1.05 (±5.44)	19.74 (±72.64)	9.8 (±41.75)	Infections and inflammation of the respiratory system without complex diagnosis, without extremely severe complication or comorbidity or a complication or comorbidity, age> 0 years, except for para/quadriplegia, without complex treatment in multidrug-resistant pathogens
E69B	119.7 (±294.1)	17.73 (±26.8)	23.82 (±64.17)	5.4 (±23.05)	45.83 (±216.58)	1.16 (±5.64)	17.71 (±67.1)	8.06 (±37.2)	Bronchitis and bronchial asthma, more than 1 day of treatment Age >55 years or with extremely severe or severe complication or comorbidity, age> 0 years or 1 day of treatment or without extremely severe or severe complication or comorbidity, age <1 year or flexible bronchoscopy, age <16 years or determined moderate treatment, with respiratory syncytial virus -Infection.
E65C	118.16 (±305.49)	17.55 (±26.44)	22.31 (±60.58)	5.01 (±22.25)	52.43 (±236.89)	1.07 (±5.26)	13.81 (±59.69)	5.99 (±33.64)	Chronic obstructive pulmonary disease without extremely severe complication or comorbidity, without complicated diagnosis, without a complication or comorbidity, age> 1 year, without specific moderately complex/expensive treatment
F62D	118.02 (±294.47)	19.78 (±27.27)	26.13 (±66.73)	5.45 (±22.51)	47.51 (±224.68)	1.34 (±6.23)	13.17 (±54.41)	4.65 (±29.04)	Cardiac insufficiency and shock without extremely serious complications or comorbidity or without dialysis, without complicated diagnosis, without complicated constellation, without specific high-level treatment, 1 day of occupancy
F48Z	117.29 (±306.82)	16.38 (±25.83)	21.75 (±60.33)	5.33 (±22.92)	49.39 (±228.94)	1.12 (±5.16)	16.21 (±66.46)	7.11 (±37.28)	Geriatric early rehabilitative complex treatment for diseases and disorders of the circulatory system
L60D	116.7 (±303.54)	15.31 (±25.14)	20.61 (±58.06)	5.06 (±22.43)	47.02 (±227.12)	1.06 (±5.52)	18.57 (±68.74)	9.07 (±40.54)	Renal insufficiency, more than one occupancy day, without dialysis, without extremely severe complications or comorbidity, age >17 years or without severe complications or comorbidity, without complex intensive care treatment > 196 /184/—expense points
K62B	116.04 (±297.88)	15.53 (±26.01)	20.84 (±59.98)	5.01 (±22.36)	45.37 (±220.28)	1.03 (±5.43)	19.12 (±69.42)	9.15 (±39.06)	Various metabolic diseases in paraplegia/tetraplegia or with complicated diagnosis or endoscopic insertion of a gastric balloon or age <16 years, one occupancy day or without extremely severe complications or comorbidity or without certain costly/highly complex treatment
F62B	109.72 (±287.12)	16.44 (±25.74)	21.88 (±60.96)	4.86 (±21.43)	44.74 (±216.62)	1.06 (±5.3)	14.26 (±59.62)	6.48 (±34.42)	Cardiac insufficiency and shock with extremely serious complications or comorbidity, with dialysis or complicated diagnosis or with certain high-level treatment or without complicated constellation, without specific high-level treatment, more than 1 day of occupancy in certain acute renal failure with extremely severe complications or comorbidity

**Table 4 ijerph-18-06669-t004:** Association between covariates and total annual costs (2014) for dental services.

Covariate	Mean	Lower Confidence Interval	Upper Confidence Interval	Description
Intercept	469.22	468.32	470.13	-
Age (years)	−4.26	−4.27	−4.25	-
Mecklenburg-Western Pommerania (ref.: Berlin)	−50.80	−50.94	−50.65	-
Brandenburg (ref.: Berlin)	−46.40	−46.54	−46.26	-
Other (ref.: Berlin)	−37.40	−37.86	−36.95	-
Social hardship status (yes, ref.: no)	13.57	13.45	13.68	-
Sex (male, ref.: female)	4.31	4.17	4.45	-
Died during follow-up (yes, ref.: no)	−52.32	−52.47	−52.17	-
G-DRG_A90A	3.62	2.91	4.32	Partial stationary geriatric complex treatment
G-DRG_B70B	−11.24	−11.90	−10.58	Apoplexy with complex neurological treatment of acute stroke
G-DRG_B80Z	9.38	8.73	10.03	Other head injuries
G-DRG_C08B	−4.34	−5.10	−3.57	Extracapsular extraction of the lens (ECCE) without congenital malformation of the lens or certain interventions on the lens
G-DRG_E65C	−24.93	−25.45	−24.41	Chronic obstructive pulmonary disease without extremely severe complication or comorbidity, without complicated diagnosis, without specific moderately complex/expensive treatment
G-DRG_E69B	−1.73	−2.43	−1.03	Bronchitis and bronchial asthma, more than 1 day of treatment Age >55 years or with extremely severe or severe complication or comorbidity, age> 0 years or 1 day of treatment or without extremely severe or severe complication or comorbidity, age <1 year or flexible bronchoscopy, age <16 years or determined moderate treatment, with RS virus -Infection.
G-DRG_E77I	−3.10	−3.50	−2.70	Infections and inflammation of the respiratory system without complex diagnosis, without extremely severe complication or comorbidity or a complication or comorbidity, age> 0 years, except for para/quadriplegia, without complex treatment in multidrug-resistant pathogens
G-DRG_F48Z	−8.91	−9.46	−8.36	Geriatric early rehabilitative complex treatment for diseases and disorders of the circulatory system
G-DRG_F49G	3.80	3.17	4.44	Invasive cardiological diagnosis except in acute myocardial infarction, without extremely severe complication or comorbidity, age >17 years, without cardiac mapping, without severe complication or comorbidity at day of treatment >1, without complex diagnosis, without specific intervention
G-DRG_F62B	−8.19	−8.46	−7.91	Cardiac insufficiency and shock with extremely serious complications or comorbidity, with dialysis or complicated diagnosis or with certain high-level treatment or without complicated constellation, without specific high-level treatment, more than 1 day of occupancy in certain acute renal failure with extremely severe complications or comorbidity
G-DRG_F62D	0.00	0.00	0.00	Cardiac insufficiency and shock without extremely serious complications or comorbidity or without dialysis, without complicated diagnosis, without complicated constellation, without specific high-level treatment, 1 day of occupancy
G-DRG_F67D	−7.04	−7.55	−6.53	Hypertension without complicated diagnosis, without extremely severe or severe complications or comorbidity, without certain moderately complex/complicated treatment, age >17 years
G-DRG_F71B	−2.70	−3.21	−2.18	Non-severe cardiac arrhythmias and conduction disturbances without extremely severe complications or comorbidity or occupancy day, without catheter-assisted electrophysiological examination of the heart, without specific high-level treatment
G-DRG_F73Z	−6.13	−6.66	−5.59	Syncope and collapse
G-DRG_G67B	−0.15	−0.63	0.32	Esophagitis, gastroenteritis, gastrointestinal bleeding, ulcer disease and various diseases of the digestive organs with other complicating factors or with extremely severe complications or comorbidity
G-DRG_G67C	1.11	0.70	1.52	Esophagitis, gastroenteritis, gastrointestinal hemorrhage, ulcer disease and various diseases of the digestive organs without certain or other complicating factors, without extremely severe complications or comorbidity
G-DRG_I34Z	−24.33	−24.94	−23.71	Geriatric early rehabilitative complex treatment with specific operating room procedure for diseases and disorders of the musculoskeletal system and connective tissue
G-DRG_I41Z	−11.60	−12.12	−11.09	Geriatric early rehabilitative complex treatment for diseases and disorders of the musculoskeletal system and connective tissue
G-DRG_I47B	8.89	8.21	9.57	Revision or replacement of the hip joint without certain complicated factors, with complex diagnosis of the pelvis/thigh, with certain endoprosthetic or joint plastic surgery of the hip joint, with implantation or replacement of a radius head prosthesis.
G-DRG_I68D	−6.78	−7.30	−6.27	Non-surgically treated diseases and injuries of the spinal column, more than one occupancy day or other femoral fracture, without sacrum fracture, without certain moderately elaborate, elaborate or highly elaborate treatment
G-DRG_J65Z	8.75	8.02	9.49	Injury of the skin, subcutis and mamma
G-DRG_K62B	−1.40	−1.87	−0.94	Various metabolic diseases in paraplegia/tetraplegia or with complicated diagnosis or endoscopic insertion of a gastric balloon or age <16 years, one occupancy day or without extremely severe complications or comorbidity or without certain costly/highly complex treatment
G-DRG_L60D	5.35	4.62	6.09	Renal insufficiency, more than one occupancy day, without dialysis, without extremely severe complications or comorbidity, age >17 years or without severe complications or comorbidity, without complex intensive care treatment >196/184/—expense points
G-DRG_L63F	−5.50	−6.03	−4.98	Infections of the urinary organs without extremely severe complications or comorbidity, without certain moderately costly/elaborate/highly costly treatment, without complex treatment multi-resistant pathogens (MRE), without certain serious infections, age >5 and <18 years, without severe complications or comorbidity or age >17 and <90 years
G-DRG_L64A	−13.17	−14.03	−12.31	Other urinary organ diseases with extremely severe or severe complications or comorbidity or certain diagnosis, more than one occupancy day or urethra-cystoscopy, congenital malformation or age <3 years
ICD_E11.90	−15.63	−15.74	−15.51	Endocrine, nutritional and metabolic diseases
ICD_E78.0	5.12	4.99	5.26	Endocrine, nutritional and metabolic diseases
ICD_E78.5	0.69	0.56	0.81	Endocrine, nutritional and metabolic diseases
ICD_E79.0	−5.43	−5.58	−5.27	Endocrine, nutritional and metabolic diseases
ICD_F03	11.17	11.03	11.31	Mental and behavioural disorders
ICD_H26.9	3.96	3.82	4.11	Diseases of the eye and adnex
ICD_H52.0	2.40	2.20	2.59	Diseases of the eye and adnex
ICD_H52.2	9.07	8.88	9.25	Diseases of the eye and adnex
ICD_H52.4	13.55	13.37	13.74	Diseases of the eye and adnex
ICD_I10.00	5.86	5.73	5.99	Diseases of the circulatory system
ICD_I10.90	4.41	4.28	4.53	Diseases of the circulatory system
ICD_I25.9	−3.66	−3.77	−3.54	Diseases of the circulatory system
ICD_I50.9	−6.35	−6.49	−6.22	Diseases of the circulatory system
ICD_I70.9	2.42	2.27	2.56	Diseases of the circulatory system
ICD_I83.9	8.99	8.84	9.15	Diseases of the circulatory system
ICD_M16.9	4.66	4.51	4.80	Diseases of the musculoskeletal system and connective tissue
ICD_M17.9	6.83	6.70	6.96	Diseases of the musculoskeletal system and connective tissue
ICD_M81.99	11.38	11.23	11.52	Diseases of the musculoskeletal system and connective tissue
ICD_N40	21.53	21.34	21.71	Diseases of the genitourinary system
ICD_R32	7.18	7.04	7.32	Symptoms, signs and abnormal clinical and laboratory findings, not elsewhere classified
ICD_R52.2	14.70	14.55	14.84	Symptoms, signs and abnormal clinical and laboratory findings, not elsewhere classified
ICD_UUU	20.62	20.48	20.75	-
ICD_Z25.1	14.77	14.66	14.88	Factors influencing health status and contact with health services
ICD_Z92.1	−0.02	−0.17	0.13	Factors influencing health status and contact with health services
ICD_Z96.1	14.54	14.39	14.69	Factors influencing health status and contact with health services

## Data Availability

Data used in this study cannot be made available by the authors given data protection rules, but may be requested at the GEWiNO.

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
