# Peer review of "Costs for Statutorily Insured Dental Services in Older Germans 2012–2017"

_ijerph, 2021, doi:10.3390/ijerph18126669_

Round 1

Reviewer 1 Report

It is very interested study. I would recommend to replace old age term by geriatric.

The result of your study shows that cost decrease by increasing in age and that costs are more in males than females, can you please expand more in these points in your discussion and to justify these result.

Another result that prosthetic and operative have the same cost, this is surprising result, can you please explain why?

Implant treatment was't mentioned in the results, can you please justify why.

Author Response

Your comment: It is very interested study. I would recommend to replace old age term by geriatric.

Our response: To remain consistent with our previous publications, we sticked to the term, but added geriatric.

Your comment: The result of your study shows that cost decrease by increasing in age and that costs are more in males than females, can you please expand more in these points in your discussion and to justify these result.

Our response: This was done.

Your comment: Another result that prosthetic and operative have the same cost, this is surprising result, can you please explain why?

Our response: We also added that.

Your comment: Implant treatment was't mentioned in the results, can you please justify why.

Our response: Implants are not covered statutorily and hence not reflected in our cost estimations. We highlight that point.

Reviewer 2 Report

The paper is well-written and addressed an important topic.

Few minor issues:

- which category of expenses included the placement of one dental implant?

- how the results you obtained could be generalized to a wider population? To all Germans, for example

- I suggest presenting some results in graphs instead of tables, for sake of clarity

- You should more detail about the novelty of the information provided and about their external validity, with particular regard to the importance that the study might have for programming informative campaign and for programming health programs.

Author Response

Your comment: Which category of expenses included the placement of one dental implant?

Our response: As implants are not covered by the statutory insurance, costs for implants are not reflected by our estimates. We highlight this once more.

Your comment: How the results you obtained could be generalized to a wider population? To all Germans, for example

Our response: This was expanded on.

Your comment:  I suggest presenting some results in graphs instead of tables, for sake of clarity

Our response: We decided to leave the tables and figures as they are, as this is in line with previous publications.

Your comment:  You should more detail about the novelty of the information provided and about their external validity, with particular regard to the importance that the study might have for programming informative campaign and for programming health programs.

Our comments: This was expanded on, thanks!